# The Positive Effects of *Poria cocos* Extract on Quality of Sleep in Insomnia Rat Models

**DOI:** 10.3390/ijerph19116629

**Published:** 2022-05-29

**Authors:** Hyeyun Kim, Injune Park, Kyunyong Park, Seohyun Park, Yeong In Kim, Byong-Gon Park

**Affiliations:** 1The Convergence Institute of Healthcare and Medical Science, Department of Neurology, International St. Mary’s Hospital, Catholic Kwandong University, Gangneung 25601, Korea; imkhy77@gmail.com (H.K.); nuyikim@ish.ac.kr (Y.I.K.); 2Department of Emergency Medicine, Gangneung Asan Hospital, Ulsan University, Gangneung 44610, Korea; myungtamjung@gmail.com; 3Department of Physiology, College of Medicine, Catholic Kwandong University, Gangneung 25601, Korea; kyunyongp@gmail.com (K.P.); quteseo77@naver.com (S.P.)

**Keywords:** *Poria cocos*, insomnia, sleep disturbance, GABA_A_ receptor

## Abstract

Sleep disorders may have various causes and can incur mental and/or physical symptoms, and affect an individual’s quality of life. In this study, we confirm that the *Poria cocos* extract (PCET) can improve sleep quality and structure by promoting inhibitory neurotransmission via the γ-aminobutyric acid (GABA) type A (GABA_A_) receptors based on the mechanisms revealed in the experiment with superior cervical ganglion neurons. Pentobarbital-induced sleep tests were conducted in order to determine whether the PCET extract improves the sleep quality and structure in normal ICR mice. Sleep latency and duration were checked with the righting reflex. To simulate the state of awakening as well as a normal sleep state, caffeine was administered orally before the PCET diet. After oral gavage of PCET, sleep latency was decreased, and total sleep duration was increased in normal and caffeine-induced sleep disturbance state. In the ACTH-induced sleep disturbed models, administration of PCET significantly reduced the sleep latency and increased the non-REM sleep duration, which was analyzed in real-time EEG by implanting wireless electrodes in SD rats. PCET was found to improve the sleep quality under a normal sleep state through the GABA_A_ receptor; it also promoted and improved the sleep quality and sleep structure in both the arousal activation state and stress-based sleep disturbance.

## 1. Introduction

The COVID-19 outbreak has extremely shortened the exposure time to healthy sunlight during daytime due to social isolation [1]. This induced a circadian rhythm disturbance and poor quality of sleep among the general population [2]. The main function of sleep is restoration with physiological and psychological resting during night sleep. Someone who has poor sleep quality may complain of lethargy, daytime somnolence, and negative emotions, such as depression/anxiety during the day, which affects their work or academic routine [1]. After the COVID19 outbreak, our society, which has become more sleepless, needs to pay more attention to healthy sleep [2].

Sleep is a complex biological process in which metabolism and motor activity are reduced while a person rests in an unconscious state. Sleep consists of rapid eye movement (REM) sleep and non-REM sleep, in which non-REM sleep consists of three stages of sleep, from stage 1 to stage 3, according to the degree of slow waves, which are subsequently observed using an electroencephalogram (EEG). Non-REM and REM sleep periods alternate during the night, with an average of 3–4 cycles, in which each cycle lasts about 90 min [3].

The causes of sleep disorders are varied and may not only incur mental and/or physical symptoms but also may affect an individual’s socioeconomic wellbeing and quality of life [4]. Primary sleep disorders are generally caused by endogenous disturbances, whereas secondary sleep disorders are typically caused by medical or neuropsychiatric conditions, such as depression, thyroid diseases, strokes, dementia, and pain [5]. Sleep disorders include insomnia, hypersomnia/narcolepsy, sleep breathing disorders, and parasomnia, such as sleepwalking and REM sleep behavior disorders [6,7]. Sleep disorders are often accompanied by neurological impairments such as strokes, dementia, neurodegenerative diseases, headaches, and various pain disorders. They could also be caused by endocrine abnormalities such as hyperthyroidism, pregnancy, menopause, diabetes, and a vitamin D deficiency [8,9]. Sleep disorders can also be induced by antidepressants that are used to treat mental disorders such as depression, anxiety disorders, phobias, and panic disorders [10]. Patients with insomnia often complain about sleep initiation or maintenance problems and that when waking up earlier than expected in the morning, they are unable to go back to sleep [11].

In common cases, γ-aminobutyric acid GABAergic, serotonergic, histaminergic, and orexinergic neurotransmitter systems in the central nervous system are targets in insomnia treatment studies [12]. GABA acts as an inhibitor in the mammalian central nervous system; it also acts on ionotropic and metabotropic receptors to activate Cl^−^ influx into neurons to hyperpolarize the membrane voltage [13]. The GABA receptor has a pentameric structure, and four subtypes have been reported depending on the action of agonists and antagonists [14]. Among GABA receptors, the subtype A receptor (GABA_A_) is an ionic receptor that has binding sites for benzodiazepines and barbiturates [15]. It is recognized as a major target for anti-anxiety, sedative, anticonvulsant, and muscle relaxation drugs [16], in which benzodiazepine drugs act as allosteric modulators on the GABA_A_ receptors to enhance the Cl^−^ influx by GABA.

*Bokryeong* (*Poria cocos*) is a sclerotium fungus that grows on the root of a pine tree; it is an herbal medicine used in northeast Asian countries for its diuretic and sedative effects [17]. *Poria cocos* has also been traditionally used for edema, sputum, palpitations, and insomnia [18]. *Poria cocos* contains parchymic acid, pinicolic acid, and 3β-hyderoxy-lanosta-7.9 as its main components, with poricoic acid, a triterpene, and its derivatives also having been isolated from *Poria cocos* and its skin [19,20]. In a more recent study, the effect of *Poria cocos* is found to appear through the GABA receptor [21], though a specific relationship has not been identified. Importantly, there have been no systematic studies conducted that investigate the improvement of sleep disorders by using a *Poria cocos* extract (PCET).

This study investigates whether a PCET, which has both sedative and insomnia relief effects, increases the Cl^−^ influx through GABA_A_ receptors and Cl^−^ currents by GABA in superior cervical ganglion (SCG) cells, as they are all known to express GABA receptor subtypes. In addition, we assess whether the extract improved the sleep quality and sleep structure, the caffeine-induced arousal-stimulating state, and other sleep disorders by using an animal model based on an endocrine disturbance induced by adrenocortical stimulating hormone (ACTH). In addition, the dosage of the extract that could induce sleep quality improvement and sleep structure in normal and sleep disorder animal models is determined in order to provide basic data for potential clinical applications.

## 2. Materials and Methods

### 2.1. Drugs Used and Laboratory Animals

#### 2.1.1. *Poria cocos* Extract

PCET was obtained by culturing the test strain, a registered variety of the National Seed Resources (Bokryeong No. 1, KMCC03342). It was extracted using 75% ethanol for 4 h before being concentrated and then freeze-dried prior to use.

#### 2.1.2. Drugs Used in the Experiment

For the sleep induction tests, 100 mg of pentobarbital from Hanlim Pharmaceuticals (Seocho, Seoul, Korea) was used. The collagenase type D, trypsin, and DNase type I used to isolate the superior cervical ganglion cells, *N*-(6-methoxyquinolyl) acetoethyl ester (MQAE) used to measure the Cl^−^ influx, and GABA used to measure the Cl^−^ currents, caffeine, and ACTH used to induce a sleep disturbance were purchased from Sigma-Aldrich (St. Louis, MO, USA).

#### 2.1.3. Laboratory Animals

The mice used in this study were 7-week-old male IcrTacSam:ICR (ICR) mice having a body weight of 20–23 g, and the rats used to construct the ACTH-induced sleep disorder model were 8-week-old male NTacSam:SD (Sprague Dawley; SD) rats having a body weight of 180–190 g. All experimental animals were purchased from Samtaco (Sungnam, Gyeonggi, Korea) and were used after acclimatization to a day and night cycle, at a temperature of 23 ± 1 °C and a humidity of 50 ± 5% at intervals of 12 h for 1 week. Rats and mice were fed a synthetic diet, and water was provided ad libitum. PCET was administered by oral gavage. All animal experiments were performed in accordance with the National Institute of Health (NIH) regulations on the protection and use of laboratory animals, and this study was performed with the approval of the Animal Experimental Ethics Committee of the Catholic Kwandong University (approval number: 2020-002). Among the three R principles of animal study, to comply with the principle of Refinement, anesthesia procedures were performed using zoletil (50 mg/kg) and meloxicam (3 mg/kg) to alleviate animal pain during the experimental procedure.

#### 2.1.4. Sleep Induction Test and Sleep Disorder Animal Model Development

In the pentobarbital-induced sleep test shown in Figure 1, 7-week-old ICR mice were stabilized for 1 week and then fasted for 24 h at 8 weeks of age. Then, 45 min after administration of PCET at concentrations of 10, 20, 40, 80, 160, and 320 mg/kg body weight (BW), sleep was induced using pentobarbital at a sleep-inducing concentration (42 mg/kg). After inducing sleep, sleep latency and sleep time were measured. The sleep latency was measured until movement ceased, and sleep time was measured as the time it took for the ICR mouse in a supine position to perform a righting reflex, i.e., the time it takes to return to a prone position on all fours after waking. The control and the dietary group for each PCET concentration were conducted with 10 mice, respectively. The control group was administrated 0.9% normal saline by oral gavage in the same volume as the PCET administration group.

In the caffeine-induced sleep disturbance animal model, the sleep-enhancing efficacy evaluated through the inhibitory action of arousal by PCET was used, the same method as for the pentobarbital sleep induction test. Caffeine was administered immediately before the administration of PCET by oral gavage. (Figure 2). Caffeine was administered at concentrations of 12.5, 25, 50, and 100 mg/kg BW to confirm the appropriate concentration of the arousal-stimulating effect of caffeine, as well as to determine the optimal concentration for increasing the sleep latency and decreasing the sleep time. The control and the caffeine group for each caffeine concentration were conducted with 10 mice, respectively. The control group was administrated 0.9% normal saline by oral gavage in the same volume as the PCET administration group.

An experiment to evaluate the sleep-enhancing efficacy of PCET in an animal model of ACTH-induced sleep disorder was conducted by stabilizing 10-week-old SD rats for 1 week and then implanting a Pinnacle wireless EEG into the rat skull (Figure 3). The detailed model construction process was conducted according to the manual provided by Pinnacle. This implant consisted of two EEG and one EMG electrode model, according to Pinnacle’s standard method (Oregon, Lawrence, KS, USA). After fixing SD rats in a stereotaxic apparatus for small animals, the skull skin was incised in the rostro-caudal direction, and a small hole was made in the skull using a small drill in order to implant the recording electrodes (Figure 3A). After implanting the EMG electrode (Figure 3B) into the neck muscle, dental acrylic was applied to insulate the EEG electrodes, and the head mount was then connected (Figure 3C). Next, the floor base used for mounting the Bluetooth device was placed, which was then fixed using dental acrylic. For real-time EEG recording, the wireless rat 2 EEG/1 EMG system (Pinnacle Technol. Oregon, Lawrence, KS, USA; 8200-K9-SL) was used. Signal amplification and transmission through the EEG electrode and Bluetooth wireless amplifier planted in the rat’s skull were received via a BT5 dongle (8274-D) connected to an IBM computer. The received EEG signals were recorded through the SIRENIA Acquisition Program (v2.1.0, Lawrence, KS, USA) with a sampling rate of 256 samples per second. Real-time EEGs recorded on the IBM computer used SIRENIA Sleep Pro software to confirm the state of awake, non-REM sleep, and REM sleep.

The sleep disorder model for the SD rats was constructed by subcutaneously injecting ACTH at a concentration of 400 μg/kg BW on days 1, 5, and 10 at 13 weeks of age, after first recovering for 2 weeks after the attachment of the Pinnacle wireless EEG electrodes. To determine whether the sleep disturbance was induced by ACTH, the sleep latency, total sleep time, non-REM sleep, and time spent awake were all measured. The efficacy of the ACTH-induced sleep disturbance due to the administration was also measured (Figure 4). All animal tests were recorded by video, and the sleep latency and sleep time were then analyzed. The control and the ACTH-injecting group were conducted with 3 rats each. The control group was subcutaneously injected with 0.9% normal saline in the same volume as the ACTH injecting group.

### 2.2. Separation of the Superior Cervical Ganglion

SD rats (200–250 g) were anesthetized by intraperitoneal injection of Zoletil (50 mg/kg). The superior cervical ganglion in the common carotid artery branch was then removed and transferred to Hank’s balanced salt solution (GibcoBRL, Sarasota, FL, USA) at 4 °C. After removing the myelin sheath and making a small gap using fine scissors, it was treated in a modified Earle’s balanced salt solution (EBSS, pH 7.4, GibcoBRL) containing collagenase (0.7 mg/mL), trypsin (0.1 mg/mL), and DNase type I (0.1 mg/mL) for 50 min [22]. After the enzyme treatment, the neurons were separated by shaking, and were then centrifuged for 10 min at 1500 rpm. After resuspending the isolated neurons in DMEM (GibcoBRL), which contained fetal calf serum (GibcoBRL; 10%) and penicillin-streptomycin (Sigma; 1%), they were used for the Cl^−^ influx test before being attached to a glass plate coated with poly-L-lysine (Sigma) in order to measure the GABA-induced Cl^−^ current.

### 2.3. Cl^−^ Influx Test and GABA-Induced Cl^−^ Current Measurement

The Cl^−^ influx in the superior cervical ganglion cells was measured using MQAE, a Cl^−^ selective fluorescent dye [23]. A buffer solution containing HPO_4_^2−^ (2.4 mM), H_2_PO_4_^−^ (0.6 mM), HEPES (10 mM), D-glucose (10 mM), and MgSO4 (1 mM) was used for the Cl^−^ influx test. After the isolated superior cervical ganglion cells were treated with 10 mM MQAE for 12 h, they were excited at 320 nm in a FlexStation 3 reader (Molecular Device, San Jose, CA, USA). Then, using a wavelength emitted at 460 nm, F_0_ and F_1_ were measured according to the presence or absence of Cl^−^, and the ratio was calculated.

The ion current was measured using a typical whole-cell patch-clamp method via a patch-clamp amplifier (EPC9, Instrutech Corp., Holliston, MA, USA) [24]. For the whole-cell current measurement, a borosilicate glass capillary was used (outer diameter: 1.65 mm; inner diameter: 1.2 mm; Corning 7052, Garner Glass Co., Claremont, CA, USA), and for single-channel recording, a filament containing a borosilicate glass capillary was used (outer diameter: 1.5 mm; inner diameter: 0.86 mm; Sutter Instrument Co., Novato, CA, USA).

All electrodes were manufactured by pulling using a P-97 Flaming-Brown micropipette puller (Sutter Instrument Co., USA). The electrode was coated with Sylgard 184 (Dow Corning, Midland, MI, USA). For the whole-cell current measurement, a resistance of 1.5 mΩ to 2.5 mΩ was used when the electrode was filled with solution. The cell-attached cover glass was then placed on an inverted microscope, and the extracellular fluid was perfused by gravity at a rate of 1 mL/min to 2 mL/min. To record the voltage clamp, the cell membrane capacitance and series resistance were corrected by more than 80%. When measuring the GABA current (I_GABA_) during the experiment, the sampling rate was 0.5 kHz, and the low-pass filter was 2 kHz (−3 dB; 8-pole Bessel filter).

All experimental results were stored in an IBM computer using Pulse/Pulsefit (v8.50) software (Heka Elektronik, Lambrecht, Germany) and then analyzed. All experiments were conducted at room temperature (21–24 °C).

### 2.4. Statistical Analysis

All experimental values were expressed as mean ± standard error, and statistical significance was confirmed using a student *t*-test (unpaired) using the Prizm software (GraphPad, Sandiego, CA, USA; v5.01). A significant difference was considered when *p* < 0.001.

## 3. Results

### 3.1. Strengthening of GABA Receptor Actions in the Superior Cervical Ganglion of Poria cocos Extract

#### 3.1.1. Increased Cl^−^ Influx Induced by *Poria cocos* Extract in Superior Cervical Ganglion Cells

To verify the efficacy of the Cl^−^ influx through the GABA_A_ receptor of PCET in the isolated superior cervical ganglion cells, the superior cervical ganglion cells were exposed to MQAE, a Cl^−^ selective fluorescent dye, for 12 h, and then excited at 380 nm in a FlexStation 3 reader.

The concentrations of Cl^−^ in the superior cervical ganglion cells were then compared by measuring the intensity of the wavelength emitted at 460 nm. As a positive control, pentobarbital (10 μM), a known agonist of the GABA_A_ receptor, was used. Compared to the negative control, the intracellular Cl^−^ influx significantly increased (Figure 5A).

PCET was treated at concentrations of 10, 30, 100, and 300 μg/mL to verify the efficacy of facilitating Cl^−^ influx into the superior cervical ganglion cells. PCET was found to increase the intracellular Cl^−^ influx in a concentration-dependent manner and, when compared to the negative control, showed a statistically significant increase at a concentration of 30 μg/mL or more (Figure 5A).

#### 3.1.2. Enhancement of GABA-Induced Cl^−^ Current by *Poria cocos* Extract in Superior Cervical Ganglion Cells

Since GABA_A_ and GABA_B_ receptors have both been expressed in superior cervical ganglion cells [25], superior cervical ganglion cells were fixed at −80 mV, and the Cl^−^ current was induced by treatment with GABA (100 μM). It was then confirmed that the GABA-induced Cl^−^ current increased when treated with PCET (1 mg/mL, 3 mg/mL). In Figure 5B, the Cl^−^ current was induced by the GABA (100 μM) treatment, and it was confirmed that the Cl^−^ current increased according to the treatment concentration of PCET (1 mg/mL, 3 mg/mL).

### 3.2. Evaluation of Sleep-Promoting Efficacy of Poria cocos Extract

#### 3.2.1. Evaluation of Sleep-Promoting Efficacy of *Poria cocos* Extract in Normal Mice

To verify whether PCET increases the inhibitory action of GABA under normal conditions, it was applied to a normal mouse (20–25 g) model, and a pentobarbital-induced sleep test was performed. Mice were maintained in a fasting state for 24 h prior to the tests. Then, 45 min after the administration of PCET at various concentrations (10, 20, 40, 80, 160, 320 mg/kg) by oral gavage, pentobarbital was injected intraperitoneally. Immediately after injection, mice were transferred to independent cages, and their sleep latency (time from injection to sleep, when the mice ceased to move) was measured. After inducing sleep, the position of the mice was changed from prone to supine. When the sleep time was over, the mice changed position from supine to prone; this righting reflex was the measure of the end of the sleep time.

As a positive control, muscimol (0.2 mg/kg), a GABA_A_ receptor agonist in poisonous mushrooms of the genus Ammonium, was applied. Compared to the negative control, the sleep latency was significantly reduced (Figure 6A), and the total sleep time significantly increased (Figure 6B). Compared to the negative control group, PCET reduced the sleep latency in a concentration-dependent manner at a concentration of 40 mg/kg or more (Figure 6A) and increased the sleep duration (Figure 6B). Concentrations of 160 mg/kg or more of the administered PCET showed a similar efficacy to muscimol, the positive control, and also showed a decrease in the sleep latency and an increase in the sleep time.

#### 3.2.2. Evaluation of Arousal Inhibitory Effect of *Poria cocos* Extract in Caffeine-Induced Arousal-Stimulating Mice

The effects of reducing the sleep latency and increasing the total sleep time of PCET under normal conditions were found to be related to the arousal inhibitory effects in the caffeine-induced arousal-stimulating ICR mouse model. First, to confirm the caffeine dose that increases sleep latency and decreases sleep time, by stimulating arousal in ICR mice, 12.5, 25, 50, and 100 mg/kg BW of caffeine were administered by oral gavage. After 45 min, pentobarbital (42 mg/kg) was injected intraperitoneally, and the sleep latency and sleep time were measured by noting both the time of motion cessation and then the subsequent righting reflex.

In Figure 7, at a dose of 25 mg/kg or more of caffeine, the arousal-stimulating effect is shown to increase the sleep latency (Figure 7A) and decrease the sleep time (Figure 7B). Caffeine at 50 mg/kg had the greatest effect on increasing the sleep latency and decreasing the sleep time, so 50 mg/kg of caffeine was selected for the caffeine-induced arousal effect.

After maintaining ICR mice in a fasting state for 24 h, caffeine (50 mg/kg) and PCET (40 mg/kg, 80 mg/kg, 160 mg/kg) were orally administered. After 45 min, pentobarbital (42 mg/kg) was injected intraperitoneally, and sleep was induced. The sleep latency increased by the caffeine injection consequently decreased in a concentration-dependent manner by the administration of PCET (Figure 7C). The total sleep time that was previously decreased by the caffeine injection was then significantly increased by the administration of PCET (Figure 7D).

### 3.3. Evaluation of PCET Sleep Improvement Effect on ACTH-Induced Sleep Disorder Model

#### 3.3.1. Inducing Sleep Disorder Using Subcutaneous ACTH Injection

To analyze the real-time awakening and sleep periods by measuring the brain waves in SD rats, we developed a model by planting a Pinnacle wireless EEG electrode into a rat skull (Figure 3). ACTH (400 μg/kg) was subcutaneously injected into the wireless EEG rat models on days 1, 5, and 10. Sleep, based on the sleep-awakening cycle, was measured in real time after the hypodermal ACTH injection in order to determine the severity of sleep disorder and the non-REM sleep time in terms of the total sleep time.

As we repeated the subcutaneous ACTH injections, the sleep latency (73.8 ± 6.3 min) was significantly decreased compared to the control group (35.6 ± 5.7 min) on day 10, whereas the non-REM sleep time increased significantly compared to the control group (Figure 8C). There was no difference in REM sleep time between the groups (Figure 8D).

#### 3.3.2. Sleep Structure Improvement Effect of PCET in the ACTH-Induced Sleep Disorder Model

At 10 days after the ACTH subcutaneous injection, we confirmed that the sleep latency and non-REM sleep time in the PCET administration group increased significantly compared to the control group. In Figure 9, the increase in sleep latency and the reduction in non-REM sleep time induced by ACTH were found to be inhibited by the oral administration of PCET. The sleep latency decreased significantly in the 40 mg/kg administration group (74 ± 4.2) compared to the control group (82 ± 5.6) and decreased remarkably in the 160 mg/kg administration group (44 ± 5.1 min) (Figure 9A). The non-REM sleep time increased significantly in the 160 mg/kg administration group (384 ± 7.1) compared to the control group (351 ± 8.3 min) (Figure 9B).

## 4. Discussion

Insomnia is a common disease, with a prevalence of around 10–30% being reported [26]. In addition, social isolation and circadian changes caused by the COVID-19 outbreak are further causative factors in the etiology of the rapidly increasing insomnia and poor sleep quality [1]. Basic and clinical studies related to insomnia are continuously underway, but it remains difficult to choose an ideal drug for safety and effectiveness in targeting the different groups of insomnia patients [2]. For the purpose of searching for sleeping pills that can be safely chosen for insomniacs, we reviewed oriental traditional herbs that have been considered to help improve insomnia. Oriental herbs were sometimes used as food ingredients and sometimes as treatments for certain diseases. In addition, for promoting sleep quality, anxiolytic/anti-depressive effects, and sedation, herbal medicines have been used in northeast Asia for centuries [27]. Recent reports have found that *Poria cocos* has diuretic, sedation, and cardiorespiratory relief effects. *Poria cocos* has long been used as a food ingredient like other herbs in the East, so it is worth research as a safe and effective treatment for insomnia. It also contains large amounts of triterpene structured compounds, including PCET [18,20]. In more recent reports, PCET has been shown to promote sleep by increasing the activity of GABA_A_ receptors [16].

In this study, we confirmed that PCET controls the inflow of Cl^−^ into neurons through the GABA receptor, verifying that PCET improves sleep latency and sleep duration in normal and sleep disturbance animal models. We also determined the optimal dosage for sleep improvement. It is known that both GABA_A_ and GABA_B_ receptors are expressed in the superior cervical ganglion [25]. We found that PCET promotes the influx of Cl^−^ and increases GABA-induced Cl^−^ current. These findings subsequently suggest that PCET may directly or indirectly stimulate GABA receptors that are important to the Cl^−^ inflow. Indeed, in pentobarbital-induced sleep tests, the administration of PCET during normal sleep led to a decrease in sleep latency and an increase in sleep duration in mice. A decrease in sleep latency can be expected to further result in rapid sleep onset and an increase in the total sleep time.

Insomnia can be induced by caffeine in coffee [28]. Caffeine is a psychoactive drug that induces awakening by inhibiting adenosine A_2_ receptors [29]. As a result, an intake of too much coffee can cause sleep disturbance [30]. In this study, as one model of a sleep disorder, caffeine was administered in mice by oral gavage at different concentrations to identify the optimal dosage (50 mg/kg) and demonstrate the maximum effects of increased sleep latency and reduced total sleep time.

Administration of caffeine to mice revealed a significant increase in sleep latency compared to the control group. The total sleep time was also significantly reduced in the caffeine administration compared to the control group. We then used additional pentobarbital-induced sleep tests to confirm the sleep promotion effect of PCET in mice with caffeine-induced sleep disturbance. The administration of over 40 mg/kg of PCET revealed a decrease in sleep latency and an increase in the total sleep duration. These results suggested that the administration of PCET can inhibit sleep disturbance induced by the arousal effects of caffeine.

EEGs are a useful tool for diagnosing brain diseases or sleep/wake cycles [31]. We implanted EEG electrodes in the skull of an SD rat and measured the EEG waves in real time because EEG waves in rats can be measured only for a limited time after inducing sleep or anesthesia [32]. Real-time EEG waves were measured by implanting electrodes in the skull of SD rats and connecting wires that transmit EEGs to a computer. This method has typical problems of line rupture and signal loss. To avoid the limitation of wire EEG, we constructed a real-time Bluetooth wireless system to obtain long-term EEG measurements in SD rats.

Insomnia can also be caused by excessive cortisol secretion by the activation of the hypothalamus–pituitary–adrenal (HPA) axis [33]. In this study, we induced the hypersecretion of cortisol by using hypodermic ACTH injections. Whether or not the subcutaneous ACTH injections caused sleep disturbance was verified 10 days after injections. Overall, the sleep latency increased, the non-REM sleep time decreased, and the wake time significantly increased. As a result, we confirmed that the subcutaneous ACTH injections induced sleep disturbance, and then administered PCET by oral gavage to confirm the decrease in sleep latency and increase in non-REM sleep time. A statistically significant decrease in sleep latency was ultimately found in the 80 mg/kg or higher administration of PCET, and an increase in non-REM sleep time was also found at doses in the range of 160 mg/kg or higher.

These results have shown that PCET significantly improved sleep quality and induced an improvement in the sleep structure of normal sleep disturbance by arousal effects of caffeine and insomnia via HPA axis activation. Subsequent attempts to identify the precise PCET concentration for improving the sleep quality and sleep structure in animal models will provide a basis for conducting clinical trials on humans. As such, further studies on the active site and efficacy of GABA_A_ receptors concerning the PCET action mechanism are necessary.

## 5. Conclusions

In this study, administration of PCET by oral gavage confirmed the effects of decreasing the sleep latency, increasing the sleep duration, and increasing the slow-wave sleep in normal state, sleep disturbance due to arousal effects of caffeine, and insomnia by HPA axis activation via ACTH subcutaneous injection. The concentration-dependent administration of PCET decreased the sleep latency and increased the sleep duration. In the ACTH-induced sleep disorder model, an increase in slow-wave sleep and a decrease in sleep latency were revealed. Overall, the results shown in this study will provide basic information as to the dosage of extracts available in the “Improvement Effect of Normal Sleep of PCET” clinical trials and will be used as foundational data for future research.

## Figures and Tables

**Figure 1 ijerph-19-06629-f001:**
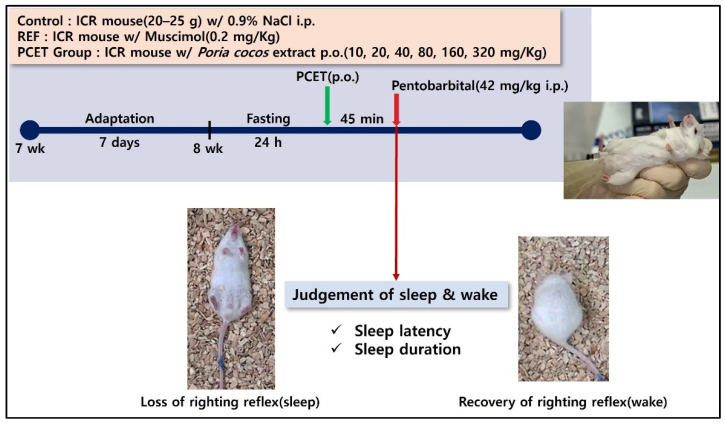
Pentobarbital-induced sleep test concept diagram. Measurement of sleep latency and sleep duration according to PCET oral intake concentration in sleep-induced mice using different intraperitoneal pentobarbital injections.

**Figure 2 ijerph-19-06629-f002:**
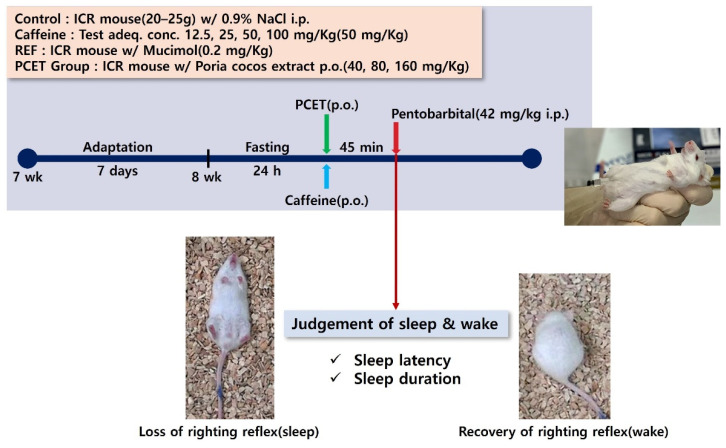
Conceptual diagram for evaluating the arousal-inhibiting efficacy of PCET in caffeine-induced arousal animal models. Measurement of sleep latency and sleep duration according to the oral intake of caffeine and PCET at different concentrations in mice that were sleep-induced by the intraperitoneal injection of pentobarbital.

**Figure 3 ijerph-19-06629-f003:**
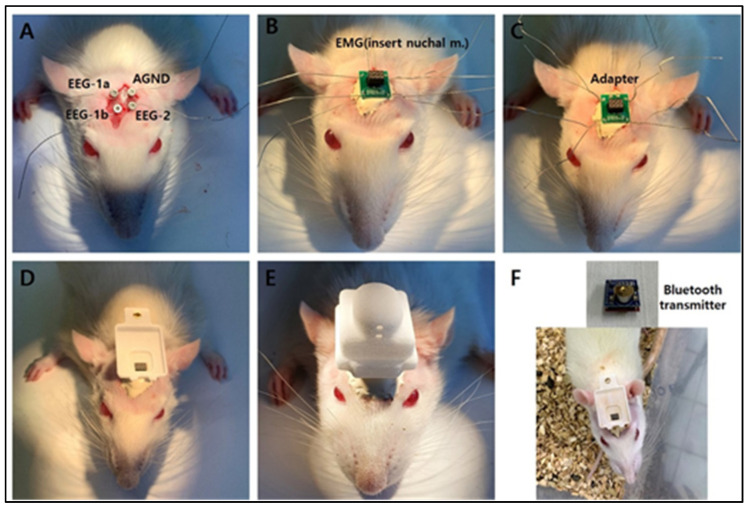
Building a continuous brainwave measuring animal model using a Pinnacle wireless EEG. (**A**) Planting Pinnacle wireless electroencephalography (EEG) electrodes in a rat skull, using two brainwave electrodes and one electromyogram electrode. (**B**) After planting the electromyogram electrodes onto a cervical muscle, dental acrylic was applied as insulation between the EEG electrodes. (**C**) Connecting the head mount. (**D**–**F**) Fixation of the bottom base that is used to mount the Bluetooth device using dental acrylic.

**Figure 4 ijerph-19-06629-f004:**
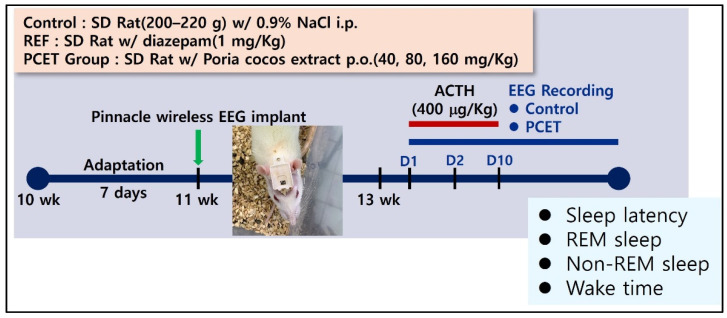
Conceptual diagram of developing ACTH-induced sleep disorder animal models. Measurement of sleep latency, REM/non-REM sleep time, and awakening time according to the intake concentration of PCET through the EEG monitor in ACTH-induced sleep disorder rats.

**Figure 5 ijerph-19-06629-f005:**
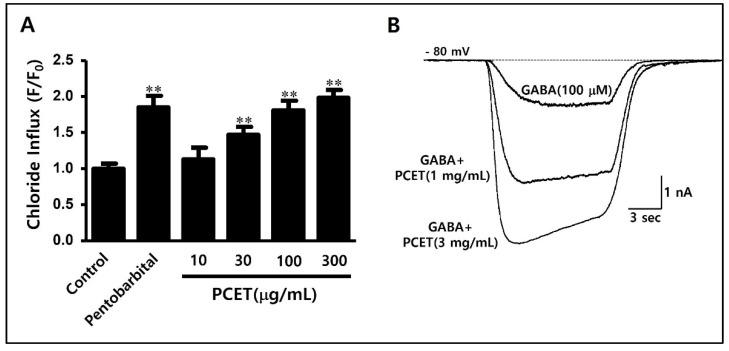
Cl^−^ influx and GABA-induced Cl^−^ current-enhancing effects of PCET in superior cervical ganglion cells. (**A**) After 12 h of exposure to MQAE, Cl^−^ selective fluorescence dye, superior cervical ganglion cells were filtrated at 380 nm in a FlexStation 3 reader and measured wavelengths emitted at 460 nm. Pentobarbital (10 μM) was used as a positive control, and the negative control was measured with no processing. PCET was administered 1 h before measurement, at a concentration range of 10–300 μg/mL. The data are shown as mean ± standard deviation. Experiments were repeated 10 times. Statistical significance was evaluated using the *p*-value, and the difference from the negative control was marked as ** *p* < 0.001. PCET increased the Cl^−^ inflow into cells concentration-dependently. (**B**) The GABA-induced Cl^−^ current strengthening effect was verified by treating 1 mg/mL and 3 mg/mL PCET on the Cl^−^ current induced using a 10 s GABA (100 μM) treatment for a superior cervical ganglion cell fixed at −80 mV. The Cl^−^ current increased according to the PCET treatment concentration.

**Figure 6 ijerph-19-06629-f006:**
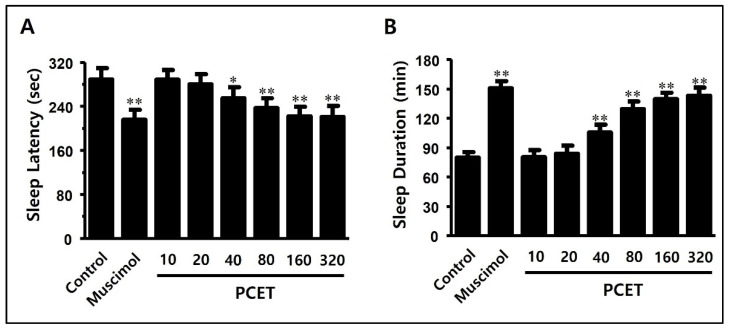
Effect of PCET on pentobarbital-induced sleep latency and sleep duration in normal mice. PCET was administered by oral gavage at concentrations of 10, 20, 40, 80, 160, 320 mg/kg BW 45 min before the intraperitoneal injection (42 mg/kg) of pentobarbital. Muscimol (0.2 mg/kg), the GABA_A_ receptor agonist, was injected intraperitoneally as a positive control. In the pentobarbital-induced sleep test, (**A**) PCET reduced the sleep latency concentration-dependently, and (**B**) increased the total sleep time. The data are marked as mean ± standard deviation. Each group consisted of 10 mice. Statistical significance was evaluated based on the *p*-value, and significant differences from the negative control were marked by * *p* < 0.01 and ** *p* < 0.001.

**Figure 7 ijerph-19-06629-f007:**
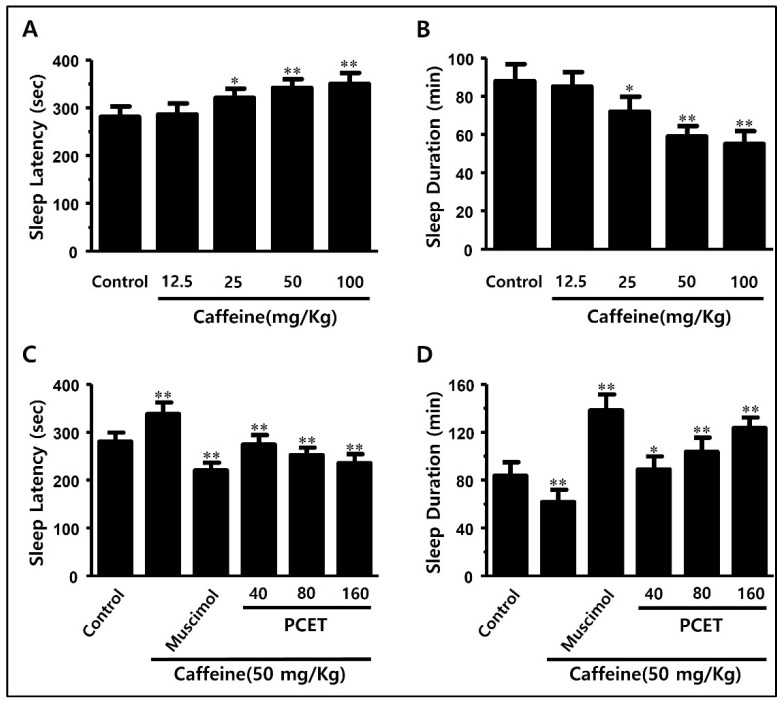
Effects of PCET on caffeine-induced sleep disturbance in mice. (**A**,**B**) Optimal caffeine concentration for inducing arousal-stimulating sleep disorders: (**A**) the effect of caffeine dosage on sleep latency and (**B**) total sleep time in pentobarbital-induced sleep tests. Caffeine (12.5, 25, 50, 100 mg/kg) was administered by oral gavage 45 min prior to the intraperitoneal pentobarbital injection (42 mg/kg). (**C**,**D**) Sleep improvement effect of PCET in caffeine-induced sleep disorders. Effect of PCET dosage on (**C**) changes in sleep latency and (**D**) total sleep time due to caffeine-induced awakening in pentobarbital-induced sleep tests. PCET was administered by oral gavage at concentrations of 40, 80, 160 mg/kg 45 min prior to the intraperitoneal pentobarbital injection (42 mg/kg), and caffeine (50 mg/kg) was also administered by oral gavage before the administration of PCET. Muscimol (0.2 mg/kg), the GABA receptor agonist, was injected intraperitoneally as a control. The data are shown as mean ± standard deviation. Each group consists of 10 mice. Statistical significance was evaluated based on the *p*-value, and significant differences from the negative control are marked by * *p* < 0.01 and ** *p* < 0.001.

**Figure 8 ijerph-19-06629-f008:**
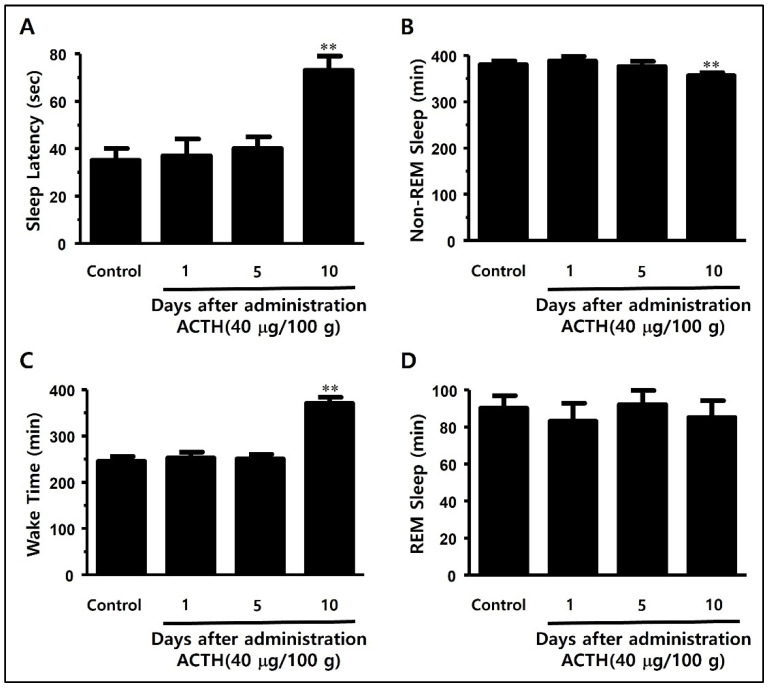
Changes in (**A**) waking time, (**B**) sleep latency, (**C**) non-REM sleep time, and (**D**) REM sleep time according to the days after ACTH injection in SD rat models. (**A**) Analysis of the change in waking time after 1, 5, and 10 days of ACTH injection shows a significant increase in waking time on day 10 compared to the control group. (**B**) Analysis of the change in sleep latency shows a significant increase in sleep latency on day 10 compared to the control group. (**C**) Analysis of the change in total non-REM sleep time shows a significant decrease in non-REM sleep time at day 10 compared to the control group. (**D**) The change in total REM sleep time shows no statistical significance. ACTH (400 μg/kg) was intraperitoneally injected on days 1, 5, and 10. The data are shown as mean ± standard deviation. Each group consists of three mice. Statistical significance was evaluated based on the *p*-value, and significant differences from the control group are marked as ** *p* < 0.001.

**Figure 9 ijerph-19-06629-f009:**
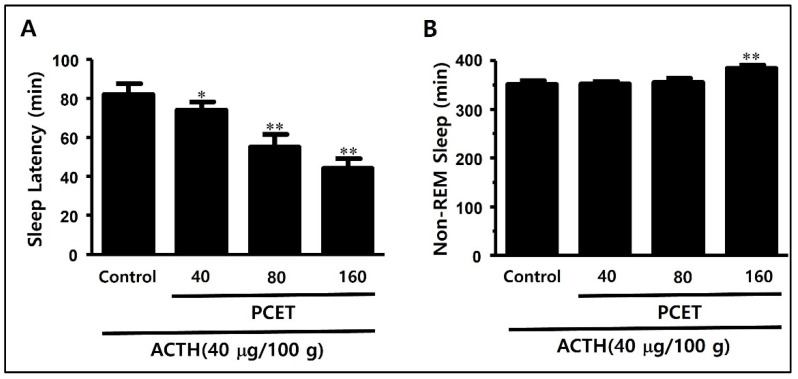
PCET effect on improving sleep disorders by subcutaneous ACTH injection. The subcutaneous injections of ACTH (400 μg/kg) in SD rats on days 1, 5, and 10, followed by oral gavage of PCET (40, 80, 160 mg/kg), resulted in (**A**) a significant concentration-dependent decrease in sleep latency, and (**B**) a significant increase in non-REM sleep time at maximum dosage (160 mg/kg). The data are shown as mean ± standard deviation. Each group consists of three rats. The statistical significance was evaluated based on the *p*-value, and * *p* < 0.01 and ** *p* < 0.001 indicate significant differences from the control.

## Data Availability

Not applicable.

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
