# Peer review of "The Positive Effects of Poria cocos Extract on Quality of Sleep in Insomnia Rat Models"

_ijerph, 2022, doi:10.3390/ijerph19116629_

Round 1
Reviewer 1 Report
In this (pilot?) study, the authors artificially induced sleeping disorders in mice and rats thanks to caffein. They then gave some of them Poria cocos extracts, and assess the quality of their sleep, while the other (control group) did not receive any medication. The authors find a positive effect of Poria cocos extracts on sleep quality.
First, I would like to commend the authors for the difficult task of setting up an EEG experiment with animals. However, I cannot endorse the publication of the manuscript for the point below:
- Description of the laboratory animals should be improved: How many subjects? Why mice and rats? Why specifically this type of mice and rats?
- A randomized design would have been better for this type of experiment, enabling the use of permutation testing between your two experimental conditions.
- Description of the EEG setup should be extended: hardware/software filtering, artifact removal, ...
Author Response
Dear Reviewer
We appreciate your review comments. We also agree with the limitations of this study. However, this study is valuable as a novel study that verified the sleep promotion effect of Poria cocos extracts in normal sleep, caffeine-induced and adrenocorticotropic hormone (ACTH)-induced sleep disturbance states. In addition, English editing for language, grammar, spelling, and overall style proposed by reviewers were conducted through Voctree, a certified localization and editing company. Please find a “point-by-point reply” to the questions raised by your opinion, and we have added a “revised manuscript”.
- Reviewer’s comments;
In this (pilot?) study, the authors artificially induced sleeping disorders in mice and rats thanks to caffein. They then gave some of them Poria cocos extracts, and assess the quality of their sleep, while the other (control group) did not receive any medication. The authors find a positive effect of Poria cocos extracts on sleep quality.
First, I would like to commend the authors for the difficult task of setting up an EEG experiment with animals. However, I cannot endorse the publication of the manuscript for the point below:
Description of the laboratory animals should be improved: How many subjects? Why mice and rats? Why specifically this type of mice and rats?
- Reply to the reviewer
I am so sorry that the experimental method did not accurately display the subjects per experimental group. However, the subjects of the experimental group was specified in the figure legends showing the experimental results. In the normal and the caffeine-induced sleep disturbance state, pentobarbital-induced sleep tests were conducted with 10 mice in control and oral diet of PCET (10, 20, 40, 80, 160, 320 mg/kg). According to the reviewer's comments, the number of subjects using the pentobarbital-induced sleep tests was specified in the methodology (2.1.4. Sleep induction test category).
Among the mice used in general experimental studies, IcrTacSam;ICR mice were used in this study. ICR mice have excellent fertility and fast growth rates, and are widely used in toxicology, pharmacology, and stability studies. This study used the ICR mouse because it was a drug test on the sleep promotion efficacy of PCET.
Excluding artificial effects, implant for wireless EEG electrodes and transmitters in skull were required for analysis of sleep quality and structure through constant EEG measurement, so rats with a larger skull size than mice were used. Among several species, NTacSam:SD (Sprague Dawley) rats were used in this study. SD rat is widely used in safety, medicinal effect, reproduction, aging, nutrition, toxicology, oncology, and pharmacological research. In general, SD species were verified with NIH before Wistar species, and the stability of scientific research was verified.
- Reviewer’s comments;
A randomized design would have been better for this type of experiment, enabling the use of permutation testing between your two experimental conditions.
- Reply to the reviewer
We verified the statistical significance between the control group and the experimental group through data analysis in Prizm software (GraphPad, v5.01). As the reviewer pointed out, randomized design could show a better verification system, but statistical significance could be confirmed even when direct comparisons between control groups and several experimental groups were directly compared in Przm software. We will try to analyze the data by the reviewer's suggestions.
- Reviewer’s comments;
Description of the EEG setup should be extended: hardware/software filtering, artifact removal, ...
- Reply to the reviewer
We apologize for not specifying a detailed description of the Wireless EEG system, data acquisition, and sleep structure analysis in the manuscript. The Wireless EEG recording system was built by implanting Pinnacle's 2 EEG/1 EMG system into the rat skull, as shown in the schematic diagram and legend in Figure 3. The detailed model construction process was conducted according to the manual provided by Pinnacle. For real-time EEG recording, the wireless rat 2 EEG/1 EMG system (Pinnacle Technol. Oregon, Lawrence, KS, USA; 8200-K9-SL) was used. Signal amplification and transmission through the EEG electrode and Bluetooth wireless amplifier planted in the rat's skull were received via a BT5 dongle (8274-D) connected to an IBM computer. The received EEG signals were recorded through the SIRENIA Acquisition Program (v2.1.0) with a sampling rate of 256 samples per second. Real-time EEGs recorded on the IBM computer used SIRENIA Sleep Pro software to confirm the state of awake, non-REM sleep, and REM sleep.
The description of the Wireless EEG system was added to the 2.1.4 category (Sleep induction test...) in Materials and Methods section.
Finally, although our explanation or correction of the reviewer's comment is not sufficient, we have revised and added the reviewer's opinion as much as possible. Once again, We sincerely thank the reviewer for giving us a critical comment to be reborn as good scientific research.

Reviewer 2 Report
This is an interesting manuscript that needs several corrections:
"Caffeine was administered orally immediately before the administration of the PCET using a sonde (Figure 2). Caffeine was administered at concentrations of 12.5 mg/kg, 25 mg/kg, 50 mg/kg, and 100 mg/kg in"
What is an administration using a sonde? It is a gavage?
Why were used these doses of caffeine?
Contents between lines 145 to 155, were obtained from some bibliographic references?
What humane endpoints were evaluated in this research?
How many animals were used?
Did all animals survive?
What is the bell? Line 100.
What software was used to perform statistical analyses?
The authors should clarify my comments in the manuscript.
Author Response
Dear Reviewer
We appreciate your review comments. We also agree with the limitations of this study. However, this study is valuable as a novel study that verified the sleep promotion effect of Poria cocos extracts in normal sleep, caffeine-induced and adrenocorticotropic hormone (ACTH)-induced sleep disturbance states. In addition, English editing for language, grammar, spelling, and overall style proposed by reviewers were conducted through Voctree, a certified localization and editing company. Please find a “point-by-point reply” to the questions raised by your opinion, and we have added a “revised manuscript”.
- Reviewer’s comments;
This is an interesting manuscript that needs several corrections:
"Caffeine was administered orally immediately before the administration of the PCET using a sonde (Figure 2). Caffeine was administered at concentrations of 12.5 mg/kg, 25 mg/kg, 50 mg/kg, and 100 mg/kg in"
What is an administration using a sonde? It is a gavage?
- Reply to the reviewer
Oral zonde is a tool used in oral diet in animal testing. You can check the shape and specification of the oral zonde on the website link (http://www.labmarket.co.kr/product/detail.html?product_no=2137&cate_no=46&display_group=1). In this study, In this study, different specifications were used in mice (17 G, 80 mm) and rats (15 G, 120 mm) for caffeine and PCET diets. It was described as sonde due to the spelling error of Oral zonde, which was corrected in the manuscript.
- Reviewer’s comments;
Why were used these doses of caffeine?
- Reply to the reviewer
Caffeine is well known for its arousal action through inhibition of adenosine A2 receptors, and numerous studies have conducted arousal tests at a concentration of 10 to 100 mg/kg body weights (BW). In this study, the strength of the sleep inhibition was tested with a concentration gradient of 2 folds the concentration range of 12.5 to 100 mg/kg BW, and as a result, a concentration of 50 mg/kg BW was selected with statistical significance of increased sleep latency and decreased sleep duration.
- Reviewer’s comments;
Contents between lines 145 to 155, were obtained from some bibliographic references?
- Reply to the reviewer
Lines 145 to 155 pointed out by the reviewer are probably confirmed to be about the production of wireless EEG models, real-time EEG recording, and sleep analysis. The Wireless EEG rat model was built using the standard method of Pinnacle Technology (Oregon, Lawrence, KS, USA).
The Wireless EEG recording system was built by implanting Pinnacle's 2 EEG/1 EMG system into the rat skull, as shown in the schematic diagram and legend in Figure 3. The detailed model construction process was conducted according to the manual provided by Pinnacle. For real-time EEG recording, the wireless rat 2 EEG/1 EMG system (Pinnacle Technol. Oregon, Lawrence, KS, USA; 8200-K9-SL) was used. Signal amplification and transmission through the EEG electrode and Bluetooth wireless amplifier planted in the rat's skull were received via a BT5 dongle (8274-D) connected to an IBM computer. The received EEG signals were recorded through the SIRENIA Acquisition Program (v2.1.0) with a sampling rate of 256 samples per second. Real-time EEGs recorded on the IBM computer used SIRENIA Sleep Pro software to confirm the state of awake, non-REM sleep, and REM sleep.
The description of the Wireless EEG system was added to the 2.1.4 category (Sleep induction test...) in Materials and Methods section.
- Reviewer’s comments;
What humane endpoints were evaluated in this research?
- Reply to the reviewer
In the animal experiments conducted in this study, the sleep-enhancing efficacy of PCET was confirmed to have a concentration of 80 to 150 mg/kg BW, and based on this, the calculation of the applied capacity of the human body was completed. The calculation formula is briefly described based on the high concentration capacity. 150 mg/kg BW (animal nose) x 3 (animal Km)/37 (human Km) = 12.16 mg/kg BW. 12.16 mg/kg BW x 70 kg (adult standard) = 850 mg/day. Based on the calculation, small clinical trials are currently underway.
- Reviewer’s comments;
How many animals were used?
- Reply to the reviewer
I am so sorry that the experimental method did not accurately display the subjects per experimental group. However, the subjects of the experimental group was specified in the figure legends showing the experimental results. In the normal and the caffeine-induced sleep disturbance state, pentobarbital-induced sleep tests were conducted with 10 mice in control and oral diet of PCET (10, 20, 40, 80, 160, 320 mg/kg). The construction of a sleep disturbance model by subcutaneous injection of ACTH and the study of sleep promotion by PCET diet used three rats per group (control, 40, 80, 160 mg/kg BW). According to the reviewer's comments, the number of subjects using the pentobarbital-induced sleep tests and ACTH-induced sleep disturbance study was specified in the methodology (2.1.4. Sleep induction test category).
- Reviewer’s comments;
Did all animals survive?
- Reply to the reviewer
In an experiment that performed the pentobarbital-induced sleep test in the study of caffeine-induced sleep disorder, all animals lived. However, when wireless EEG implants were performed on the rat skull, six out of a total of 24 died during the procedure, and six were not included in the experimental group due to insufficient continuous EEG measurement signals.
This study was conducted under the supervision and approval of the Animal Ethics Committee for the maintenance and management of animals, and certain temperatures, humidity, diet, and water supply were conducted. All experiments were carried out in accordance with "Care and Use of Laboratory Animals" published by the US National Institutes of Health, and the study was ap-proved by the Committee for the Care and Use of Laboratory Animals at Catholic Kwandong University (2020-002). Of course, in this study, anesthesia and pain in sleep tests or wireless EEG implant in the rat skull were accompanied. We applied for an animal experiment plan under the theme of " Evaluation of sleep promotion and the effect of improving sleep quality by Poria cocos extract in the sleep disorder animal model in vivo" and made efforts to minimize animal pain by following the guidelines with deliberation and approval from three referees in “Committee for the Care and Use of Laboratory Animals at Catholic Kwandong University”.
Among the three R principles of animal study, to comply with the principle of Refinement, anesthesia procedures were performed using zoletil and meloxicam (3 mg/Kg BW) was orally administered to alleviate animal pain during the experimental procedure. The animal study in this manuscript was approved by the “Committee for the Care and Use of Laboratory Animals at Catholic Kwandong University” after deliberation on whether it complies with the 3R principle of animal study.
- Reviewer’s comments;
What is the bell? Line 100.
- Reply to the reviewer
“Bell” is a misspelling of “species”. In the manuscript, "the bell" was modified to “all experimental animal species”.
- Reviewer’s comments;
What software was used to perform statistical analyses?
- Reply to the reviewer
We verified the statistical significance between the control group and the experimental group through data analysis in Prizm software (GraphPad, v5.01).
Finally, although our explanation or correction of the reviewer's comment is not sufficient, we have revised and added the reviewer's opinion as much as possible. Once again, We sincerely thank the reviewer for giving us a critical comment to be reborn as good scientific research.

Reviewer 3 Report
The manuscript by Kim et al presents an interesting potential use of Poria cocos extract (PCET) as an agent for reducing sleep latency, increase sleep duration and, overall, helping sleep-related issues.
Their overall approach is correct and the results are interesting, but, in my view, the manuscript does show some worth-addressing problems that I'll try to enumerate.
Overall:
-Although the overall tone of the writting is formal and adequate, the reader can find numerous grammar and spelling mistakes, along with sentences that can be difficult to understand. Thus, extensive language revision is suggested.
Intro:
-All the mentioned literature seems relevant, and the research problem is sufficiently justified. It lacks, however, some explanation about the rationale behind the experiment on the Superior cervical ganglion, which is important and that, somehow, is not mentioned in the Abstract, either.
Methods:
-The authors chose good and reliable techniques for the research problem, and the pentobarbital-induced sleep model is adequate. However, it is not clear why they used one model of insomnia induction (oral caffeine administration) for one experiment and another one (ACTH injection) for other. It is highly advisable that the authors clarify this point, not only for scientific clarity and honesty but also for the reader to not get lost while going through the paper.
-Although 24h food deprivation is ethically allowed, it would be good that the authors pointed it out explicitly in the manuscript, along with the reason behind it (most likely ensuring that there are not confounding factors altering PCET and oral caffeine consumption).
-It would be good that the authors mentioned the software used for statistical analysis.
-The authors state that statistical significance is set as p<0.005 (p. 6, l. 221), but in figures it is stated that p<0.01 = * and p<0.001 = **
Results:
-The order of the different experiments composing the manuscript is different to the one presented during the Methods section, which the reader might finding confusing. In my view, the manuscript might benefit from starting with the behavioral experiments (latency to sleep + duration of sleep), then moving to the EEG study, and finally exploring the molecular determinants with the in vitro ganglion experiment.
-Did the authors note any relevant effect after PCET-treated animals woke? Did they measure any additional variable (e.g.: attention, startle response...)? I am aware that it might not have been the goal of this study, but is definitely something that the authors might want to take into account in future studies
Discussion:
-The authors pretty much succeed in wrapping up a cohesive story, but might want to keep an eye on the order by which they present the data (see the last paragraph about Results, first point).
-The incorporation of a relationship between COVID-19 outbreak-related social isolation and circadian rythms factors is relevant and grants higher impact of the manuscript, so it might benefit from incorporating it in the Intro (either just there or also).
Author Response
Dear Reviewer
We appreciate your review comments. We also agree with the limitations of this study. However, this study is valuable as a novel study that verified the sleep promotion effect of Poria cocos extracts in normal sleep, caffeine-induced and adrenocorticotropic hormone (ACTH)-induced sleep disturbance states. In addition, English editing for language, grammar, spelling, and overall style proposed by reviewers were conducted through Voctree, a certified localization and editing company. Please find a “point-by-point reply” to the questions raised by your opinion, and we have added a “revised manuscript”.
- Reviewer’s comments;
The manuscript by Kim et al presents an interesting potential use of Poria cocos extract (PCET) as an agent for reducing sleep latency, increase sleep duration and, overall, helping sleep-related issues.
Their overall approach is correct and the results are interesting, but, in my view, the manuscript does show some worth-addressing problems that I'll try to enumerate.
Overall:
-Although the overall tone of the writting is formal and adequate, the reader can find numerous grammar and spelling mistakes, along with sentences that can be difficult to understand. Thus, extensive language revision is suggested.
- Reply to the reviewer
Thank you for checking the manuscript carefully and giving an opinion on English correction. English editing for language, grammar, spelling, and overall style proposed by reviewers were conducted through Voctree, a certified localization and editing company. After professional English correction, this was reflected in the revised manuscript.
- Reviewer’s comments;
Intro:
-All the mentioned literature seems relevant, and the research problem is sufficiently justified. It lacks, however, some explanation about the rationale behind the experiment on the Superior cervical ganglion, which is important and that, somehow, is not mentioned in the Abstract, either.
- Reply to the reviewer
To confirm the effectiveness of GABA receptors in the mechanism of sleep promotion efficacy of PCET, we tried to confirm whether the Cl- currents by GABA in neurons expressing both GABAA and GABAB receptors is substantially increased by PCET. We tried to identify the GABA effectiveness of PCET in vitro by separating the superior cervical ganglion neurons, a representative neuron known to express both GABAA and GABAB receptors, using the electrophysiological patch method. Potentiation of GABA efficacy by PCET in cervical ganglion cells was added to the abstract.
- Reviewer’s comments;
Methods:
-The authors chose good and reliable techniques for the research problem, and the pentobarbital-induced sleep model is adequate. However, it is not clear why they used one model of insomnia induction (oral caffeine administration) for one experiment and another one (ACTH injection) for other. It is highly advisable that the authors clarify this point, not only for scientific clarity and honesty but also for the reader to not get lost while going through the paper.
- Reply to the reviewer
The causes of sleep disorders are known in various forms, but all sleep disorders model that reflect them could not be produced. In this study, sleep disorders induced as a cause of lifestyle were typically targeted, and the first model was sleep disorders caused by excessive caffeine intake. Caffeine promotes awakening through inhibition of adenosine A2 receptors, resulting in sleep disorders. Another model is a stress-induced sleep disorder, in which cortisol hypersecretion caused by activation of the hypothalamus-pituitary-adrenal axis (HPA axis) during chronic stress causes sleep disorders. There are extensive studies showing that HPA axis hyperactivity by chronic ACTH administration is useful for the study of sleep disorders and depression. Since research results have been widely reported that functional ingredients indicating sleep promotion efficacy alleviate caffeine-induced sleep disorders but are not effective for stress-induced sleep disorders, this study attempted to verify their validity by applying PCET formulas to both models.
- Reviewer’s comments;
-Although 24h food deprivation is ethically allowed, it would be good that the authors pointed it out explicitly in the manuscript, along with the reason behind it (most likely ensuring that there are not confounding factors altering PCET and oral caffeine consumption).
-It would be good that the authors mentioned the software used for statistical analysis.
-The authors state that statistical significance is set as p<0.005 (p. 6, l. 221), but in figures it is stated that p<0.01 = * and p<0.001 = **
- Reply to the reviewer
We verified the statistical significance between the control group and the experimental group through data analysis in Prizm software (GraphPad, v5.01). Statistical significance is correct as specified in the figure legends. The p < 0.005 specified in the methodology was modified to p < 0.001.
- Reviewer’s comments;
Results:
-The order of the different experiments composing the manuscript is different to the one presented during the Methods section, which the reader might finding confusing. In my view, the manuscript might benefit from starting with the behavioral experiments (latency to sleep + duration of sleep), then moving to the EEG study, and finally exploring the molecular determinants with the in vitro ganglion experiment.
- Reply to the reviewer
Thank you very much for the reviewer's opinion that even considers the order of experimental results described in the manuscript. As we wrote the manuscript, we also considered the format of placing the mechanism of increasing GABA effectiveness by PCET at the back of the manuscript, as in the reviewer's opinion. However, the application of PCET to behavioral studies without any description may appear to be the result of conducting the study without understanding the basis, so it was inevitably placed in the first order. We would like to appreciate you once again giving us your opinion by connecting the readers' understanding in consideration of the technical orders of manuscript.
- Reviewer’s comments;
-Did the authors note any relevant effect after PCET-treated animals woke? Did they measure any additional variable (e.g.: attention, startle response...)? I am aware that it might not have been the goal of this study, but is definitely something that the authors might want to take into account in future studies
- Reply to the reviewer
In this study, mice and rats orally receiving PCET showed no difference in unusual behavioral abnormalities or post-recovery diets compared to the control group receiving 0.9% normal saline when waking up after sleep. As the reviewer’s opinion, attention or acoustic startle reflex could not be quantitatively measured. Currently, additional studies on the sleep-promoting efficacy of PCET are underway in various situation for sleep disorders, and as the reviewer’s comments, we will make sure to check for abnormal reactions. Thank you for your opinion of the very critical points in the animal study.
- Reviewer’s comments;
Discussion:
-The authors pretty much succeed in wrapping up a cohesive story, but might want to keep an eye on the order by which they present the data (see the last paragraph about Results, first point).
- Reply to the reviewer
As you know, please refer to the explanation of the comments in the result section.
- Reviewer’s comments;
-The incorporation of a relationship between COVID-19 outbreak-related social isolation and circadian rythms factors is relevant and grants higher impact of the manuscript, so it might benefit from incorporating it in the Intro (either just there or also).
- Reply to the reviewer
We agree with your recommendation. Social issue like COVID-19 outbreak added to the intro of the introduction like below.
COVID-19 Outbreak has significantly reduced exposure to healthy sunlight during daytime due to social isolation. This leads to a circadian rhythm change and poor quality of sleep-in public. The function of sleep is restoration with physiological and psychological resting during nighttime. Someone who have poor sleep quality complain of lethargy, daytime somnolence, and negative emotion such as depression and anxiety during the day, which affects their work or academic routine. After the Covid19 Outbreak, our society, which became more sleepless, began to pay more attention to healthy sleep.
Finally, although our explanation or correction of the reviewer's comment is not sufficient, we have revised and added the reviewer's opinion as much as possible. Once again, We sincerely thank the reviewer for giving us a critical comment to be reborn as good scientific research.

Round 2
Reviewer 1 Report
The authors took seriously the comments raised during the previous round and address most of the concern.
Author Response
Dear Reviewer
We really appreciate your sincere advice. We revised a large part according to the review comments, and from the first to the final manuscript, the value as a research paper has increased in a wide range of ways, from research background, introduction, methodology, and results and discussion. Once again, I sincerely thank you.
- Reviewer’s comments;
The authors took seriously the comments raised during the previous round and address most of the concern.
- Reply to the reviewer
We made detailed revisions to improve the quality of the research paper. Thank you again for your advice.

Reviewer 2 Report
The authors performed almost all the corrections requested. However, they should correct the route of administration to gavage
https://www.ncbi.nlm.nih.gov/pmc/articles/PMC3276962/
Because in animal experiments is not zonde. This is really a big terminology mistake.
Author Response
Dear Reviewer
We really appreciate your sincere advice. We revised a large part according to the review comments, and from the first to the final manuscript, the value as a research paper has increased in a wide range of ways, from research background, introduction, methodology, and results and discussion. In addition, English editing for language, grammar, spelling, and overall style proposed by reviewers were conducted through Voctree, a certified localization and editing company. Please find a “point-by-point reply” to the comments raised by your opinion, and we have added a “certification of English editing” and a “revised manuscript”. Once again, we sincerely thank you once again for your advice.
- Reviewer’s comments;
The authors performed almost all the corrections requested. However, they should correct the route of administration to gavage.
https://www.ncbi.nlm.nih.gov/pmc/articles/PMC3276962/
Because in animal experiments is not zonde. This is really a big terminology mistake.
- Reply to the reviewer
In this study, feeding of caffeine and PCET was conducted as a one-time. Gavage may be misunderstood as a long-term gastrostomy tube, so it was described as administration using oral zonde. However, as you point out, we think that oral gavage is the right expression and terminology. We modified “oral zonde” to “oral gavage” in the revised manuscript. We thank you for letting us know the exact concept of terminology in animal study.
